# Abdominal Desmoid: Course, Severe Outcomes, and Unique Genetic Background in a Large Local Series

**DOI:** 10.3390/cancers13153673

**Published:** 2021-07-22

**Authors:** Gilad Ophir, Shamai Sivan, Strul Hana, Rosner Guy, Gluck Nathan, Fliss Isakov Naomi, Klausner Joseph, Wolf Ido, Merimsky Ofer, Goldberg Yael, Levi Zohar, Zer Alona, Kariv Revital

**Affiliations:** 1Tel-Aviv Medical Center, Department of Gastroenterology and Hepatology, Sackler Faculty of Medicine, Tel-Aviv University, Tel Aviv 6997801, Israel; hanas@tlvmc.gov.il (S.H.); guyr@tlvmc.gov.il (R.G.); nathang@tlvmc.gov.il (G.N.); naomifl@tlvmc.gov.il (F.I.N.); revitalk@tlvmc.gov.il (K.R.); 2Tel-Aviv Medical Center, Institute of Oncology, Sackler Faculty of Medicine, Tel-Aviv University, Tel Aviv 6997801, Israel; sivansh@tlvmc.gov.il (S.S.); idow@tlvmc.gov.il (W.I.); oferm@tlvmc.gov.il (M.O.); 3Tel-Aviv Medical Center, Department of Surgery, Sackler Faculty of Medicine, Tel-Aviv University, Tel Aviv 6997801, Israel; Klausner.joseph@tlvmc.gov.il; 4Rabin Medical Center, The Raphael Recanati Genetic Institute, Sackler Faculty of Medicine, Tel-Aviv University, Tel Aviv 6997801, Israel; yaelgo43@clalit.org.il; 5Rabin Medical Center, Department of Gastroenterology and Hepatology, Sackler Faculty of Medicine, Tel-Aviv University, Tel Aviv 6997801, Israel; zoharl@clalit.org.il; 6Rabin Medical Center, Institute of Oncology, Sackler Faculty of Medicine, Tel-Aviv University, Tel Aviv 6997801, Israel; alonaz@clalit.org.il

**Keywords:** desmoid tumor, familial adenomatous polyposis, next generation sequencing

## Abstract

**Simple Summary:**

Abdominal desmoids are rare fibroblastic tumors. Though these tumors do not display metastatic potential, their locally aggressive nature can cause severe outcomes. Most cases appear sporadically, but 5–15% are associated with familial adenomatous polyposis (FAP) syndrome. Current consensus recommendations do not offer a standard sequence of therapy due to the lack of data for some treatment options. Here, we present an ongoing clinical experience with abdominal desmoids. The majority of our patients suffered severe outcomes such as need for surgery or major tumor complications. A small, but unique group of 16 non-FAP mesenteric desmoid was found to harbor genetic alterations in cancer associated genes other than *APC,* including *CHEK2, BLM, ERCC5, MSH6,* and *PALB2*.

**Abstract:**

Introduction: Abdominal desmoid tumors are locally aggressive tumors that develop in familial adenomatous polyposis (FAP) patients, within the mesentery or abdominal wall. The understanding and implications of the treatment regimens are evolving. Aim: To assess the course, treatment, and outcomes of FAP and non-FAP abdominal desmoids and their related genetic alterations. Methods: Retrospective cohort study. Demographics, tumor characteristics, oncological and surgical history, complications, genetic-testing, and mortality data were retrieved from two tertiary referral centers. Results: Sixty-two patients were identified (46 FAP and 16 non-FAP). Thirty-eight patients (61.3%) underwent surgical procedures (12 urgent and 26 elective). Out of 33 tumor resections, 39.4% recurred. Hormonal therapy, COX-inhibitors, chemotherapy, imatinib, and sorafenib were used in 35 (56.4%), 30 (48.4%), 18 (29.1%), 7 (11.3%), and 8 (12.9%) of patients, respectively, with a 2 year progression-free survival of 67.8%, 57.7%, 38.4%, and 28.5%, respectively. Forty-one patients (66.1%) suffered complications: bowel obstruction (30.6%), hyperalimentation (14.5%), ureteral obstruction (12.9%), perforation (11.3%), abscess formation (3.2%), and spinal cord compression (3.2%). Non-FAP patients carried pathogenic mutations in *CHEK2, BLM, ERCC5, MSH6,* and *PALB2*. Conclusions: Abdominal desmoids are mostly FAP-related and are associated with severe outcomes. We also report a group of non-FAP abdominal desmoids, which includes patients with additional cancer-related gene alterations. This interesting group should be further explored.

## 1. Introduction

Desmoid tumors are fibroblastic lesions thought to be associated with dysregulated wound healing. They can develop at any anatomic site, though the three dominant sites are extremities, abdominal wall, and intra-abdominal [1,2,3]. Initial presentation can be as asymptomatic masses, but progressive growth and infiltration of adjacent structures may cause symptoms, pressure, and ischemia, as well as obstruction of the internal organs [4,5,6,7,8]. Despite their locally aggressive nature, these tumors have no metastatic potential. Desmoid tumors are rare, with an incidence of about 2–5 new cases per 1 million population per year [1,2,9]. Most cases appear sporadically, mainly at the trunk [10], but 5–15% are associated with familial adenomatous polyposis syndrome (FAP) [3,11], most of which are mesenteric and may present as multiple lesions [10].

Risk factors for desmoid, specifically abdominal, include high estrogen states such as pregnancy [12], and previous abdominal trauma or surgery [13,14]. As FAP patients account for only about 5–15% of all desmoid cases, a rare condition in itself, there are a paucity of data regarding this population and other heredities of this tumor. The natural history of desmoid tumors is unpredictable. Some patients experience spontaneous regression, some have stable disease for years, while others have a progressive disease that requires intervention [15]. Abdominal desmoids carry a higher risk for life threatening complications involving internal organs. FAP patients typically present with mesenteric desmoids that are more aggressive and multifocal than sporadic cases [10].

Several consensus meetings have recently attempted to standardize management, but did not address abdominal desmoids separately [1]. The latest consensus states that in asymptomatic patients, active surveillance is an acceptable strategy [1,15,16]. For symptomatic patients or those with progressively growing tumors, surgical resection is a viable option, especially for non-abdominal tumors, provided that the tumor presents in a favorable site not compromising the adjacent structures. Tumors that are unresectable, particularly mesenteric tumors in FAP patients, should be treated medically [17,18]. Medical options include a combination of hormonal anti-estrogen therapy with COX inhibitors [19], tyrosine kinase inhibitors (TKIs) [20,21], or chemotherapeutic regimens [22].

The current consensus recommendations do not offer a standard sequence of therapy because of a lack of data for some options. Thus, it has been recommended to consider the level of evidence, overall response rate, progression free survival (PFS), ease of administration, and expected toxicity of the administered drug, by following a five-dimensional model [1]. Typically, an initially less toxic treatment is followed by a more toxic one [1].

We present ongoing clinical experience with abdominal (mesenteric and abdominal wall) desmoids among FAP patients from two referral centers, as well as their course and response to treatment. We report a unique group of 16 non-FAP mesenteric desmoids, some with other cancers, and the associated findings of cancer related genetic alterations, thus suggesting additional molecular pathways related to this tumor.

## 2. Methods

This is a retrospective cohort study from two referral centers. Patients with a clinical and biopsy proven diagnosis of either mesenteric or abdominal wall desmoid tumor were identified from GI oncology, hereditary cancer, or soft-tissue tumors clinics. We included patients with an established FAP diagnosis and identified *APC* mutation. Additionally, we separately identified patients with mesenteric desmoids, in which *APC* testing did not reveal any pathogenic variants. Demographics, genetic workup, tumor characteristics collected from CT/MRI (initial and maximal size, number of tumors, and location), complications (i.e., bowel obstruction and perforation, ureter obstruction, GI bleeding, and need for total parenteral nutrition (TPN)), cancer history, surgical history, and mortality were documented. Medical therapy was recorded including COX inhibitors (celecoxib), hormonal therapy (tamoxifen), chemotherapy (either methotrexate and vinblastine or anthracycline-based), or TKI (imatinib or sorafenib).

Treatment evaluation: Failure of treatment was defined as growth of tumor or development of tumoral complications necessitating a switch to another line of treatment. We defined specific treatment duration as time from the initiation of therapy to either treatment failure, death, or last follow up. The two year PFS rate was defined as the percentage of patients on a certain treatment regimen that did not progress for 2 years. A single physician in each medical center extracted data into a structured uniform database. All FAP patients were offered genetic workup by Sanger sequencing or next generation sequencing (NGS), or were tested for a known familial mutation. Patients with mesenteric desmoid and no known FAP-related phenotype or family history underwent endoscopic evaluation, and if polyps were found, underwent genetic evaluation as above. Some non-FAP cases underwent genetic consultation because of additional tumors and underwent a multi-gene NGS panel. All of the genetic tests were performed by medically certified laboratories.

## 3. Statistical Analysis

Continuous variables are presented as median + interquartile range (IQR), and dichotomous variables as proportions. Association between categorical variables was evaluated using Pearson’s Chi-Square or Fisher’s exact test. Mann–Whitney test was used to compare the distribution of continuous variables between the study groups. *p* < 0.05 was considered statistically significant for all of the analyses. SPSS software was used for all of the analyses (IBM version 25, 2017. Armonk, NY, USA).

## 4. Results

### 4.1. Patients and Desmoid Characteristics

Sixty-two patients were identified: 46 FAP patients from 34 families, and 16 non-FAP patients. The median follow up time was 72.4 (IQR 37.1–151.9) months. Table 1 summarizes the baseline characteristics among all of the patients.

Most FAP patients (45/46) underwent bowel surgery before tumor appearance, compared with only one non-FAP patient (6.25%), who underwent a right colectomy due to an advanced polyp. Ten female patients (31.2% of females) were diagnosed during or after pregnancy, six of whom were FAP-associated. Only two FAP patients (4.3%) received prophylactic therapy with celecoxib, as there is no official prophylaxis policy at either center.

### 4.2. Genetic Findings

Twenty-nine FAP patients (63.04%) had a known pathogenic genetic variant in the *APC* gene. An additional seventeen patients were previously diagnosed with FAP, but did not have documentation of their genetic testing. Twenty-five of the FAP patients (54.3%) had a clear history of desmoid tumors in their family. In 7 of 29 patients, the mutation was located between codon 1400 and the 3’ end of the *APC* gene. We found no correlation between the location of mutation and overall complications (23.5% of patients with mutations beyond codon 1400 suffered complications, compared with 20% with mutations closer to the 5’ end, *p* > 0.999). Among the non-FAP patients, eight performed genetic workup, revealing four pathogenic variants and five variants of uncertain significance (VUS) (Table 2).

### 4.3. Desmoid Related Morbidity

A total of 39 patients (62.9%) suffered significant complications, with a median of two events per patient (IQR 1–2). Table 1 summarizes these adverse events and their distribution between FAP and non-FAP patients. We found no correlation between maximal size or number of desmoid tumors and occurrence of complications (*p* = 0.28 and 0.7, respectively). Adverse events were not noted in any of the FAP-patients with only abdominal wall tumors.

### 4.4. Surgical and Medical Treatment

Thirty-eight patients (61.3%) underwent abdominal surgery during follow up. Surgical procedures were divided into emergency procedures due to life threatening complications and elective procedures. Twelve patients (31.6%) underwent emergency procedures—six due to small bowel perforation (one of the seven cases was considered a micro-perforation and was treated conservatively), five due to small bowel obstruction, and one patient due to uncontrolled gastrointestinal bleeding. Nine of these twelve patients were FAP-associated and three were non-FAP. Twenty-six of the 38 surgical patients (68.4%) underwent elective desmoid resection—17 FAP associated and 9 non-FAP. Complete surgical resection was achieved in 33/38 cases (86.8%). Of these, 13 patients (39.4%) had tumor recurrence, 3 after emergency surgery and 10 after elective resection. Ten patients with recurrence of the tumor were FAP associated (76.9%) and three were non-FAP (23.1%).

We found no correlation between the maximal size or number of desmoid tumors and need for surgery (*p* = 0.88 and 0.34, respectively) for all types of surgery. Similar results were seen for elective and emergency surgeries, separately. We also noted no correlation between these parameters and risk for tumor recurrence (*p* = 0.64 and 0.47, respectively).

Forty-six patients (41 FAP) were treated medically (Table 3 summarizes the prevalence, duration, and outcome per medication). Only five non-FAP patients received medical therapy (one had sorafenib and four had chemotherapy). Three had progressive disease requiring surgery, and the other two remained stable.

The two-year PFS is displayed in Figure 1. Only one patient on sorafenib therapy was treated for over 2 years; therefore, we looked at the median treatment duration (6.8 months), by which only 1/9 (11.1%) patients failed sorafenib therapy. COX inhibitors and hormonal therapy were initiated early after tumor diagnosis, at a median of 4.1 and 9.1 months, respectively. These drugs were used for the longest duration (median of 32.2 and 44.1 months, respectively). Table 3 and Figure 2 illustrate the initiation and duration of all medications.

Sixteen patients (25.8%) did not receive any medical treatment throughout follow-up: twelve had mesenteric tumors, one had an abdominal wall tumor and three had both. Eleven of the sixteen (68.7%) underwent surgical resection (nine elective and two urgent), five of whom relapsed. Five patients (8.1% of the cohort) received neither medical nor surgical treatment, and were being monitored for disease progression.

### 4.5. Oncologic Outcomes and Mortality

Of the 46 FAP patients, 18 (39.1%) had other malignant or benign tumors during follow-up. Eleven patients (23.9%) had osteomas or fibromas (Gardner syndrome), six (13.1%) had papillary thyroid carcinoma, two had colorectal cancer (4.3%), one had endometrial carcinoma, and one had a neuroendocrine tumor. Four non-FAP patients (25%) had malignancies—three with renal cell carcinoma and one with germ cell tumor.

Two patients (3.2%), both with FAP related desmoid, died during follow-up. One died due to progression of a neuroendocrine tumor. The other one had an abdominal desmoid with bilateral urethral obstructions requiring bilateral nephrostomies, and recurrent bowel obstructions requiring hyperalimenation with TPN. This patient had a prolonged hospitalization with recurrent bouts of urinary tract infections, and an invasive fungal infection attributed to her central venous catheter. She eventually succumbed to urosepsis.

## 5. Discussion

In this retrospective study of 62 patients with abdominal desmoid, mostly FAP-associated and with median follow-up of 6 years, we describe grave outcomes and complexity of clinical management. Sixty-three percent of our study population developed major, typically several complications. All complications were due to mesenteric desmoids causing mainly bowel obstruction (30.6%), bowel perforation (11.3%), and ureteral obstruction (12.9%). Other studies report similar rates of bowel obstruction. Xhaja and Church described a rate of 35% among 133 mesenteric FAP-associated desmoid patients, 69% of which required surgical intervention [4]. Soravia et al. reported slightly higher rate of 58% bowel obstruction and a 22% ureteral obstruction rate. However, a lower 2% perforation rate was reported among 53 FAP-associated desmoid patients [5]. Bowel perforation as a complication of desmoid tumors is mentioned in other studies only as case reports [6,7,8]. We reported two patients with neural involvement by desmoid tumor: one with spinal cord compression and another with sciatic nerve involvement. We found only one report with a similar case of a 12 year old girl who presented with a paraspinal sporadic desmoid with intra-spinous extension of the tumor causing scoliosis and paralysis [23]. Other case reports of paraspinal desmoid tumors appearing after spinal interventions [24] did not present with neurological deficits. Our cohort presented a high recurrence rate of 39.4% after surgical removal of the tumor. Other studies, which included mainly sporadic cases, demonstrated recurrence rates between 20–53% of cases [25,26,27,28]. Our data are in the highest range of recurrence rates in the literature, with the note that our population consisted mainly of FAP patients. While the burden of morbidity in our cohort was substantial, the mortality rate was surprisingly low—only two FAP patients (5%) died during follow-up. Quintini et al. showed a significantly higher mortality rate of 22.1% in a cohort of 154 FAP patients over a similar follow up duration [29]. The lower mortality rate in our study might be attributed to dedicated desmoid clinics at both centers and to advanced therapeutic regimens that were introduced since Quintini’s study, such as TKI therapy. Other studies show low mortality rates, however only a minority of abdominal desmoids were included [19,30].

Assessing the efficacy of systemic therapy in desmoid tumors is complicated. These tumors can spontaneously regress in more than 25% of cases [16], and this might be mistaken as a response to therapy. Furthermore, mesenteric desmoids often appear as soft tissue infiltration in the mesentery, lacking the clear 3D look of a parenchymatic solid malignancy needed for Response Evaluation Criteria in Solid Tumors (RECIST). Abdominal imaging experts in our center felt that the RECIST criteria could not be applied accurately to all patients in our study. This is supported by a work from the French sarcoma group, where changes in MRI intensity signaling better predicted response to treatment when compared with the RECIST criteria [31]. We chose to use 2-year-progression free survival (PFS) to show the effect of therapy. This parameter was used by Gounder et al. in his trial on sorafenib in desmoid tumors [21]. We determined progression according to symptoms or MRI, and all patients were discussed in a multidisciplinary tumor board.

Because of small numbers and the retrospective nature of this work, we were unable to directly compare various treatment modalities, as most patients were treated with either COX inhibitors and hormonal therapy or chemotherapy as first line therapy, and chemotherapy or TKIs as second line therapy. While COX inhibitors and hormonal therapy were started the earliest and lasted for the longest durations—approximately 3 and 4 years, respectively, these were utilized in asymptomatic patients who might have benefitted from close surveillance as well. There are no comparative trials to assess the efficacy of COX inhibitors in desmoid tumors, and regression reported in retrospective studies could represent the spontaneous regression that has been previously reported for desmoids [15]. Fiore et al. described a 2-year PFS of 89.6% among patients treated with toremifene [19]. Quast et al. reported similar results with sulindac and hormonal therapy, as 85% of patients showed regression or stabilization of the tumor [32]. However, a prospective study among 59 pediatric patients receiving tamoxifen and sulindac revealed less optimal results. The response rate was only 8% and the 2-year PFS was only 36% [33].

The two-year PFS rates in our study were between 57–67% for these treatment groups, demonstrating a less optimal response than previous retrospective studies, but a much better response than that seen in the pediatric one [33]. The long duration of COX inhibitors and hormonal therapy in our cohort may support their role as a viable first-line therapy due to their moderate effectiveness and low related morbidity.

Published data regarding the efficacy of chemotherapy vary. Azzarelli et al. demonstrated a 5 years PFS of 67% using methotrexate and vinblastine [34]. Similarly, Palassini et al. demonstrated a 2-year PFS rate of about 80% using methotrexate and vinblastine [30]. This is in contrast with the results by Constantinidou et al., who showed a median time to progression of only 9 months using the same treatment regimen in 18 sporadic desmoid patients [22]. Our results are more similar to the latter study, with a 2-year PFS of about 40% and a median duration of therapy of only 18 months.

TKIs were often used as second- or third-line therapy. Our results with imatinib show a very low 2 years PFS of 28.5%, which is considerably low compared with the available data. Chugh et al. enrolled 51 desmoid patients for imatinib therapy, and demonstrated 1 and 3 years PFS of 66 and 58%, respectively [20]. Penel et al. described a 2-year PFS of 55% [35], and Kasper et al. showed progression arrest rate of 45% at 2 years [36]. These studies consisted mainly of sporadic desmoid patients.

It is premature to discuss our results regarding sorafenib, as only three patients hadbeen using it for more than 1 year. However, we report that with a median follow up of 6.8 months, only 11.1% (1/9) had a documented progression. A large double-blind case-control study of 87 desmoid patients described similar results of a 2-year PFS of 81% in the sorafenib group compared with 36% in the placebo group. The Kaplan–Meier curve showed a 9-month PFS of about 90% [21].

We describe patients with non-FAP associated mesenteric desmoids carrying cancer-predisposing genetic variants. Half of these patients, who underwent NGS multigene panels, were found to carry a pathogenic variant, and another 37.5% had a variant of unknown significance. Alterations in *CHEK2*, a tumor suppressor gene, were discovered in two patients. *CHEK2* is associated with Li-Fraumeni syndrome, an autosomal dominant disorder that manifests as multiple malignancies, including soft tissue sarcomas and breast cancer, supporting causality [37]. Other data have disproven this association [38]. Additional alterations were found in *BLM*, responsible for Bloom syndrome, an autosomal recessive disease associated with a wide range of malignancies (notably carcinomas, hematologic malignancies, and sarcomas) [39]. Three subjects were surprisingly found to carry genetic alterations associated with well-established genetic syndromes not usually related to desmoid tumors: two Lynch syndrome patients carried *MSH6* mutation and VUS in *MSH2* and one patient had a mutation in *PALB2*, which encodes a *BRCA2* interacting protein that increases susceptibility to breast and pancreatic cancers [40]. Pathogenic variants in these genes may be associated with increased risk for desmoids, however this requires further study. We therefore suggest genetic consultation and testing also for non-FAP patients with mesenteric desmoids.

The limitations of our study include its retrospective nature; the lack of a standardized treatment algorithm for desmoid patients, lack of control group of patients on active surveillance only, and missing data for referred patients. Our study also lacks a histology of surgical specimens; hence, we cannot report if complete surgical excision of the tumor, proliferation parameters, or molecular profiles had an impact on the outcomes.

The study strengths lie in the relatively large FAP-cohort of this rare condition; detailed information from two major referral centers; and a novel subgroup of non-FAP cases, some with alterations in cancer related genes that have not been previously described. We tracked details of several pharmacologic treatment lines and could evaluate them over time.

## 6. Conclusions

Abdominal desmoid tumors, while non-metastatic in nature, are associated with severe clinical outcomes. Therapeutic approaches vary, and include medical and surgical options, but with limited real-life data regarding their effectiveness because of the rarity of the disease. Further prospective studies are required to evaluate the therapeutic response and molecular profiles of desmoid tumors. We present a small, but unique, group of non-FAP mesenteric desmoids, some with genetic alterations in cancer associated genes other than APC. Cases with abdominal desmoid and no polyps, especially with background of additional tumors, should be referred for genetic consultation and appropriate testing.

## Figures and Tables

**Figure 1 cancers-13-03673-f001:**
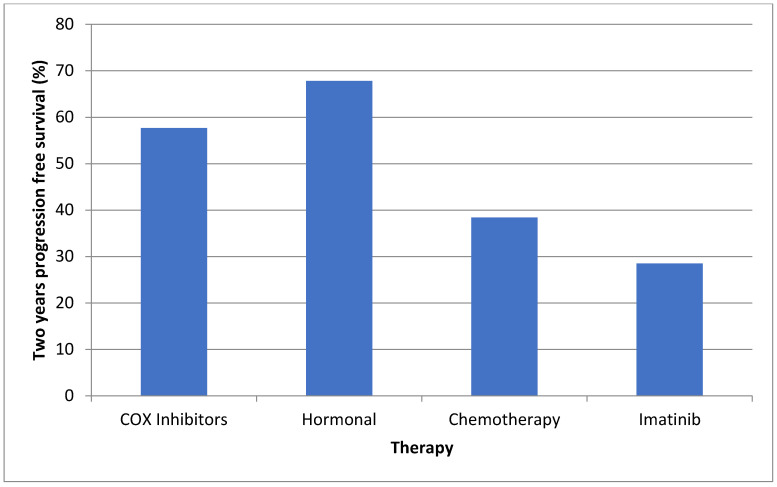
Two-year progression free survival according to different treatment modalities.

**Figure 2 cancers-13-03673-f002:**
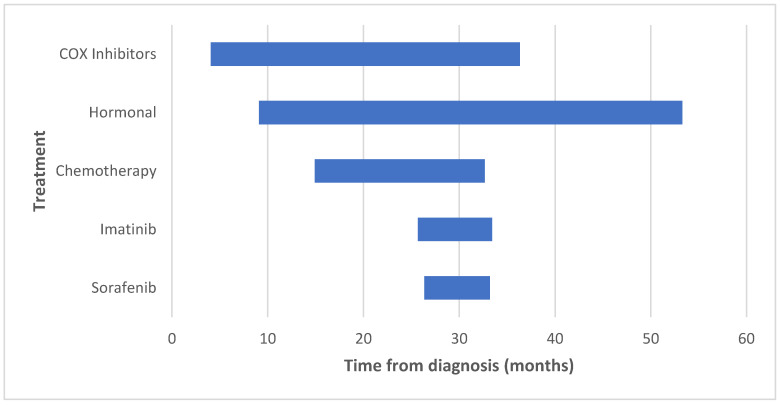
Initiation and duration of the various treatment modalities.

**Table 1 cancers-13-03673-t001:** Basic characteristics of the study population and complications among FAP and non-FAP groups.

Parameter	FAP (*n* = 46)	Non-FAP (*n* = 16)	All (*n* = 62)	*p* Value
Sex—male (%)	21 (45.6)	9 (56.2)	30 (48.3)	0.46
Median age at desmoid diagnosis (IQR)	31 (23–38)	43.5 (33–63.25)	34 (23–44)	0.007
Median follow up, months (IQR)	83.1 (53.9–170.9)	33.9 (21.9–61.5)	72.4 (37.1–151.9)	0.004
Median number of desmoids per patient (IQR)	2 (1–3)	1 (1–2.2)	2 (1–3)	0.26
Median desmoid size at diagnosis, cm (IQR)	4.7 (3.5–7.6)	8 (4.2–13)	5.5 (3.5–8.8)	0.08
Median maximal desmoid size, cm (IQR)	7.9 (4.5–10.7)	8 (5.4–14)	8 (4.7–11.5)	0.76
Desmoid location				
Abdominal wall (%)	4 (8.7)	1 (6.25)	5 (8.1)	0.13
Mesentery (%)	27 (58.7)	14 (87.5)	41 (66.1)	0.6
Both (%)	15 (32.6)	1 (6.25)	16 (25.8)	0.08
Adverse events, patients (%)	31 (67.3)	10 (62.5)	41 (66.1)	0.72
Bowel perforation (%)	5 (10.8)	2 (12.5)	7 (11.3)	>0.999
Bowel obstruction (%)	16 (34.7)	3 (18.7)	19 (30.6)	0.34
Small bowel resection (%)	13 (28.2)	6 (37.5)	19 (30.6)	0.53
Ureter obstruction (%)	7 (15.2)	1 (6.25)	8 (12.9)	0.66
Ischemic colitis (%)	2 (4.3)	0	2 (3.2)	>0.999
GI bleeding (%)	2 (4.3)	0	2 (3.2)	0.56
Need for TPN (%)	7 (15.2)	2 (12.5)	9 (14.5)	>0.999
Abscess formation (%)	1 (2.7)	1 (6.25)	2 (3.2)	>0.999
Spinal cord compression (%)	1 (2.1)	1 (6.25)	2 (3.2)	0.45
Surgery before appearance of desmoid (%)	42 (91.3)	1 (6.25)	43 (69.3)	<0.001
Pregnancy before appearance of desmoid (% of females)	6 (24)	4 (57.1)	10 (31.2)	0.26
Death (%)	2 (4.3)	0	2 (3.2)	>0.999

**Table 2 cancers-13-03673-t002:** Characteristics of non-FAP desmoid patients and their genetic alterations.

Patient Number	Sex	Age at Diagnosis	Desmoid Location	Complications	Other Malignancies	Genetic NGS Panels Performed	Genes with Pathogenic Variants	Genes with VUS
1	Male	67	Mesenteric	None	None	Yes	*CHEK2*	*APC*
2	Male	29	Mesenteric	None	Germ cell tumor	Yes	*ERCC5* *BLM*	*TP53*
3	Male	54	Mesenteric	Bowel perforation and liver abscess	None	Yes	*-*	*BLM*
4	Female	20	Mesenteric	None	None	Yes	*-*	*MSH2*
5	Male	69	Mesenteric	Bowel obstruction and resection	RCC	Yes	*-*	*CHEK2*
6	Female	38	Mesenteric	None	None	Yes	*MSH6*	-
7	Male	45	Mesenteric	Bowel obstruction and need for TPN after resection	None	Yes	*PALB2*	-
8	Male	30	Mesenteric	Bowel and ureter obstruction	RCC	Yes *	-	-
9	Female	37	Mesenteric and abdominal wall	Bowel perforation and abscess formation	None	No	-	-
10	Male	62	Abdominal wall	None	None	No	*-*	*-*
11	Male	68	Mesenteric	Small bowel resection	None	No	*-*	*-*
12	Female	34	Mesenteric	None	None	No	*-*	*-*
13	Female	81	Mesenteric	None	RCC	No	*-*	*-*
14	Male	59	Mesenteric	Need for TPN after resection	None	No	*-*	*-*
15	Female	42	Mesenteric	Small Bowel resection	None	No	*-*	*-*
16	Female	9	Mesenteric	Sciatic nerve involvement	None	No	*-*	*-*

NGS—next generation sequencing; VUS—variant of uncertain significance; * Patient 8 was found to carry an alteration in fumarate hydratase (FH) gene, which was eventually deemed non-pathogenic.

**Table 3 cancers-13-03673-t003:** Prevalence, duration, and outcome of the different treatment modalities.

Therapy	Patients Treated as 1st Line (%)	Patients Treated as 2nd Line (%)	Patients Treated as 3rd Line (%)	Patients Treated as 4th Line (%)	Treatment Initiation, Months from Diagnosis, Median (IQR)	Treatment Duration, Months from Initiation, Median (IQR)	Two Year Progression Free Survival (%)
COX 2 inhibitors	29 (46.7)	0	1 (1.6)	0	4.1 (0–33.9)	32.2 (6.6–92.9)	57.7
Hormonal	33 (53.2)	1 (1.6)	1 (1.6)	0	9.1 (0–48.7)	44.1 (7.1–103.7)	67.8
Chemotherapy	8 (12.9)	7 (11.2)	3 (4.8)	0	16.8 (5.1–49.5)	17.7 (7.9–23)	38.4
Imatinib	1 (1.6)	4 (6.4)	2 (3.2)	0	25.6 (7.7–60.8)	7.7 (6.5–20.3)	28.5
Sorafenib	1 (1.6)	6 (9.6)	1 (1.6)	1 (1.6)	26.3 (13.1–34.9)	6.8 (3.6–11.2)	N/A

## Data Availability

The data presented in this study are available upon request from the corresponding author.

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
