# Peer review of "Abdominal Desmoid: Course, Severe Outcomes, and Unique Genetic Background in a Large Local Series"

_cancers, 2021, doi:10.3390/cancers13153673_

Round 1
Reviewer 1 Report
In the manuscript titled, "ABDOMINAL DESMOID- COURSE, SEVERE OUTCOMES AND UNIQUE GENETIC BACKGROUND IN A LARGE LOCAL SERIES," the authors present a solid retrospective analysis of FAP and non-FAP desmoid tumors. These tumors are complicated to manage clinically and the additional insights that this manuscript provides is much needed in the field. The manuscript is well written, with a detailed discussion that puts this study in the proper context of other clinical management and outcome studies.
There are only two minor criticisms that can be easily addressed:
1) Spacing throughout the paper. There are several lines where the left/right justification introduces large spaces in the sentences.
2) In figure 2, the drugs are presented in the reverse order. It would be preferable to keep the order consistent with the rest of the paper.
Author Response
We thank the reviewer for this evaluation.
1) Spacing throughout the paper. There are several lines where the left/right justification introduces large spaces in the sentences).
Response: Thank you, spaces have been corrected throughout the manuscript.
2) In figure 2, the drugs are presented in the reverse order. It would be preferable to keep the order consistent with the rest of the paper.
Response: Thank you, the drug order has been changed in accordance to the reviewer comment.
Reviewer 2 Report
This manuscript is an original article that retrospectively investigated 62 patients with abdominal desmoid tumors regarding characteristics, complications, treatment and genetic testing. The authors showed clinical and genetic characteristics of abdominal desmoid tumor in FAP as well as non-FAP patients.
This study was conducted well, and the methods are appropriate. The data are presented clearly.
The results will be of interest to clinicians and researchers in the field.
However, the following major and minor issues require clarification:
Major
- The description in medical treatment seems to be insufficient. The authors stated most patients were treated with either COX inhibitors and hormonal therapy or chemotherapy as first line therapy. Therefore, the authors should assess and discuss the results with regimens including combination therapies such as COX inhibitors and hormonal therapy as well.
Minor
- (P6L170) Thirteen patients had tumor recurrence after complete resection, and the authors mentioned the high recurrence rate may be due to the predominance of FAP patients. The authors should show the number of FAP -associated patients who had tumor recurrence after surgery.
- (P8L206) Please describe the detailed cause of sepsis in the patient who died.
Author Response
We thank the reviewer for this evaluation and supply comments below.
Major
1.The description in medical treatment seems to be insufficient. The authors stated most patients were treated with either COX inhibitors and hormonal therapy or chemotherapy as first line therapy. Therefore, the authors should assess and discuss the results with regimens including combination therapies such as COX inhibitors and hormonal therapy as well.
Comment: We appreciate and thank the reviewer for this important comment.
Patients received various systemic treatments (cox-2 inhibitors, selective estrogen receptor modulators, Imatinib, Sorafenib, chemotherapy) and we outlined the duration and sequence of treatments in table 3 and figure 2.
Assessing the efficacy of systemic therapy in desmoid tumors is complicated. These tumors can spontaneously regress in more than 25% of cases (I), and this might be mistaken as a response to therapy. Furthermore, mesenteric desmoids often appear as soft tissue infiltration in the mesentery, lacking the clear 3D look of a parenchymatic solid malignancy needed for Response Evaluation Criteria in Solid Tumors (RECIST). For example in a large cohort on COX inhibitors and hormonal therapy no formal response criteria were applied (II).
We consulted with the head of abdominal imaging unit in our center (an experienced specialist in MRI) when we collected the data, and she felt that the RECIST criteria could not be applied accurately to all patients. This is supported by a work from the French sarcoma group (III), where changes in MRI intensity signaling better predicted response to treatment, when compared to RECIST criteria.
We chose to use 2-years- progression free survival (PFS) to show the effect of therapy (Figure 1). This parameter has been used by Gounder MM et al (NEJM 2018 Dec 20;379(25):2417-2428) in his trial on Sorafenib in desmoid tumors. We determined progression according to symptoms or MRI, and all patients were discussed in a multidisciplinary tumor board.
I. Bonvalot S et al. Spontaneous regression of primary abdominal wall desmoid tumors: more common than previously thought. Ann Surg Oncol 2013 Dec;20(13)4096-102
II. Quast DR, Schneider R, Burdzik E, et al. Long-term outcome of sporadic and FAP-associated desmoid tumors treated with highdose selective estrogen receptor modulators and sulindac: a single-center long-term observational study in 134 patients. Fam Cancer 2016;15:31e40
III. Crombe’ A et al. Progressive Desmoid Tumor: Radiomics Compared With Conventional Response Criteria for Predicting Progression During Systemic Therapy-A Multicenter Study by the French Sarcoma Group. AJR Am J Roentgenol 2020 Dec;215(6):1539-1548
Minor
- (P6L170) Thirteen patients had tumor recurrence after complete resection, and the authors mentioned the high recurrence rate may be due to the predominance of FAP patients. The authors should show the number of FAP -associated patients who had tumor recurrence after surgery.
Comment: We thank the reviewer for this comment. 10/13 patients with recurrence of desmoid were FAP associated. - (P8L206) Please describe the detailed cause of sepsis in the patient who died.
Comment: This patient had an abdominal desmoid with bilateral urethral obstructions requiring bilateral nephrostomies, and recurrent bowel obstructions requiring hyperalimenation with TPN. This patient had a prolonged hospitalization with recurrent bouts of urinary tract infections, and an invasive fungal infection attributed to her central venous catheter. She eventually succumbed to urosepsis
Round 2
Reviewer 2 Report
I appreciate the authors’ polite response.
I understand why assessing the efficacy of systemic therapy in desmoid tumors is complicated.
I recommend the authors include these explanations in the discussion.
Author Response
We thank the reviewer for his comments. A short paragraph explaining the complexity of desmoid tumor assessment has been added to the discussion section of the manuscript.